# FAST: A Dual-tier Few-Shot Learning Paradigm for Whole Slide Image Classification

**Kexue Fu**[16*]**, Xiaoyuan Luo**[23*]**, Linhao Qu**[23*]**, Shuo Wang**[23]**, Ying Xiong**[4]**,**
**Ilias Maglogiannis**[5]**, Longxiang Gao**[16‡]**, Manning Wang**[23‡]

[1]Key Laboratory of Computing Power Network and Information Security, Ministry of Education,
Shandong Computer Science Center (National Supercomputer Center in Jinan),
Qilu University of Technology (Shandong Academy of Sciences), Jinan, China
[2]Digital Medical Research Center, School of Basic Medical Sciences, Fudan University
[3]Shanghai Key Lab of Medical Image Computing and Computer Assisted Intervention
[4] Fudan University      [5] University of Piraeus
[6] Shandong Provincial Key Laboratory of Computing Power Internet and Service Computing,
Shandong Fundamental Research Center for Computer Science, Jinan, China
{fukx, gaolx}@sdas.org, {19111010030, mnwang}@fudan.edu.cn

## Abstract

The expensive fine-grained annotation and data scarcity have become the primary obstacles for the widespread adoption of deep learning-based Whole Slide Images (WSI) classification algorithms in clinical practice. Unlike few-shot learning methods in natural images that can leverage the labels of each image, existing few-shot WSI classification methods only utilize a small number of fine-grained labels or weakly supervised slide labels for training in order to avoid expensive fine-grained annotation. They lack sufficient mining of available WSIs, severely limiting WSI classification performance. To address the above issues, we propose a novel and efficient dual-tier few-shot learning paradigm for WSI classification, named FAST. FAST consists of a dual-level annotation strategy and a dual-branch classification framework. Firstly, to avoid expensive fine-grained annotation, we collect a very small number of WSIs at the slide level, and annotate an extremely small number of patches. Then, to fully mining the available WSIs, we use all the patches and available patch labels to build a cache branch, which utilizes the labeled patches to learn the labels of unlabeled patches and through knowledge retrieval for patch classification. In addition to the cache branch, we also construct a prior branch that includes learnable prompt vectors, using the text encoder of visual-language models for patch classification. Finally, we integrate the results from both branches to achieve WSI classification. Extensive experiments on binary and multi-class datasets demonstrate that our proposed method significantly surpasses existing few-shot classification methods and approaches the accuracy of fully supervised methods with only $0.22\%$ annotation costs. All codes and models will be publicly available on `https://github.com/fukexue/FAST`.

## 1   Introduction

With the advent of Whole Slide Images (WSI) scanners, automated diagnosis based on WSIs has become a critical problem in the field of computational pathology [53, 48, 26]. Due to the huge size of WSI [51], deep learning-based methods typically divide it into a series of patches and apply classification models to each patch individually. Fully supervised methods annotate each patch

---

*Equal Contribution      ‡Corresponding Author

38th Conference on Neural Information Processing Systems (NeurIPS 2024).

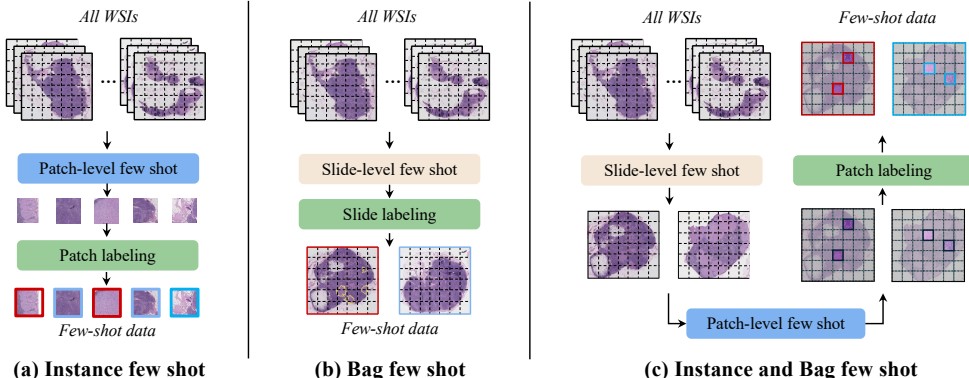

| (a) Instance few shot | (b) Bag few shot | (c) Instance and Bag few shot |

Figure 1: Different few-shot learning paradigms for WSI classification. (a) The instance few-shot method divides all WSIs into a series of patches, then selects a few samples at the patch level and annotates them at the patch level. The red box represents positive samples, and the blue box represents negative samples. (b) The bag few-shot method directly selects a few WSIs at the slide level and annotates them weakly at the slide level. (c) Our method first selects a few WSIs at the slide level, then annotates a few patches for each selected WSI. Compared to (a) and (b), our method significantly reduces annotation costs while providing patch-level supervision information.

and then train the classification model in an end-to-end manner [14, 7, 44]. However, fine-grained annotation of WSIs requires expert knowledge and is extremely expensive. To overcome these issue, weakly supervised methods formulate the WSI classification task as a multi-instance learning (MIL) problem [48, 18, 27]. In MIL, each WSI (or slide) is a bag containing thousands of instances (patches) cropped from the slide. They only use slide-level labels to train the classification model. These studies assume the availability of abundant WSIs in clinical settings. However, due to wide staining variations [30, 8], multiple cancer types [33], and rare diseases [25], many clinical scenarios can only access a limited number of WSIs. The dual obstacles of fine-grained annotation difficulties and data scarcity severely limit the available supervised information for model training. Therefore, how to avoid expensive fine-grained annotations while fully utilizing limited WSIs has become a key issue in the field of WSI classification.

Recent works [37, 61, 10, 42, 20, 54, 55, 59] in natural images have demonstrated the effectiveness of few-shot learning under limited data. Inspired by these studies, some methods [8, 6, 49, 45] gathered all patches obtained from dividing WSIs and randomly selected some patches for annotation, as shown in Figure 1(a), termed instance few shot, and then used existing few-shot learning methods from natural images for classification. These methods provide strong supervision signals for network training, but discard a large number of unlabeled patches. Our comparative experiments indicate that the methods from natural images perform poorly in few-shot WSI classification. Another methods combine weakly supervised learning with few-shot learning, such as TOP [38], which annotates only a few slide labels, as depicted in Figure 1(b), termed bag few shot. Compared to instance few shot, bag few shot methods can utilize all patches. However, slide labels belong to weak supervision signals, and the few-shot scenario leads to even greater scarcity of supervision information for MIL-based weakly supervised learning methods. Therefore, the accuracy of TOP exhibits a significant gap compared to the fully supervised learning methods [35]. Overall, the key reason for the poor performance of existing few-shot WSI classification methods is that these methods cannot simultaneously utilize strong supervision from patch labels and the remaining unlabeled patches, lacking sufficient mining of available WSIs.

In this paper, we propose a novel and efficient dual-tier **F**ew-shot learning p**A**radigm for W**S**I classifica**T**ion, named FAST. It consists of a annotation-efficient dual-level WSI annotation strategy and a parameter-efficient dual-branch WSI classification framework, which fully utilizes existing vision-language foundation models and can be rapidly applied to various WSI classification tasks.

Specifically, we first design a dual-level WSI annotation strategy, as shown in Figure 1(c). Under this strategy, a small number of WSIs are selected at the slide-level, followed by labeling a small number of patches within each selected WSI. Experts only need to annotate a very small number of patches without needing to perform fine-grained annotation on the entire WSI, significantly reducing the cost of fine-grained annotation while increasing the speed of annotation. Based on the proposed

annotation strategy, we formulate the few-shot WSI classification task as a dual-tier few-shot learning problem. Unlike conventional few-shot learning [37], which uses an "N-way K-shot" setting, FAST's "shots" consists of two levels: bag and instance. Subsequently, under the setting of dual-tier few-shot learning, we propose a dual-branch few-shot WSI classification framework that combines vision-language (V-L) models [41, 1]. To fully utilize the prior knowledge of V-L models and the limited WSIs, the classification framework includes a cache branch and a prior branch. For the cache branch, we use the image encoder of the V-L model CLIP [41] to extract features of all patches, then construct a cache model using the labeled instances, and finally classify each instance through knowledge retrieval. However, due to the very limited number of annotated instances, the cached model has only limited knowledge and lacks generalization capability. To improve the cache model's performance, we incorporate a large number of unlabeled instances from the WSIs into the cache model, and treat their labels as learnable parameters, which effectively increases the knowledge capacity of the cache model. During training, we use annotated instances as supervision to optimize these parameters. For the prior branch, we first use GPT4-V [1] to obtain task-related prompts, and then utilize CLIP's text-image matching prior and prompt-learning techniques to design a learnable visual-language classifier. Finally, we integrate the outputs of the cache and prior branches to obtain the final classification results. This framework is based on the foundational model, requires minimal optimization of parameters (cache model and prompt vectors), and can achieve parameter-efficient fine-tuning with a small amount of WSIs and labels. Additionally, by leveraging the prior knowledge of the foundational model, FAST can maintain good accuracy and generalization even with extremely limited annotated data. These characteristics make FAST suitable for rapid adaptation to various WSI classification tasks. In summary, our main contributions are as follows:

- We propose a novel few-shot learning paradigm for WSI classification, which achieves high-accuracy WSI classification and rapid adaptation to various WSI classification tasks under low-cost annotation.

- We propose an efficient dual-level WSI annotation strategy, which can provide patch-level supervisory information at a cost close to that of slide-level annotation.

- We propose a learnable cache model based on the foundation model, which fully utilizes both annotated and unannotated patches. Furthermore, we utilize the prior knowledge of vision-language foundation models to construct a visual-language classifier, combining both to further enhance the performance of the classification framework.

- Extensive experiments demonstrate that our method achieves state-of-the-art performance on the CAMELYON16 dataset and the TCGA-RENAL dataset. In addition, compared to fully supervised methods, the annotation cost is only $0.22\%$ of that.

## 2 Related Work

**WSI Classification**    According to supervised information, WSI classification methods can be divided into two categories: fully supervised learning methods based on patch-label and weakly supervised learning methods based on slide-label. Fully supervised learning methods directly draw inspiration from supervised learning methods in natural images [23, 11, 12, 15, 58, 32]. Relevant studies on diseases such as breast cancer [44, 43], lung cancer [7, 3, 31, 14], and prostate cancer [22, 29] indicate that such methods have approached or even surpassed expert diagnostic accuracy [36]. However, expensive fine-grained annotation prevents their widespread adoption in clinical practice. Weakly supervised learning methods [48, 18, 27, 39, 40, 19, 48, 28, 35] formulate the WSI classification task as a multi-instance learning problem, avoiding expensive fine-grained annotation. Although great progress has been made, these studies rely on large amounts of training data and and cannot address the common issue of data scarcity encountered in practical clinical settings. In this paper, we propose a novel WSI classification paradigm composed of an efficient annotation strategy and a prior knowledge classification framework. The proposed method not only alleviates the poor performance issues of existing methods caused by data scarcity and difficulties in fine-grained annotation, but also enables rapid adaptation to various disease WSI classification tasks.

**Few-shot Learning for WSI Classification**    Inspired by the success of few-shot learning in natural images, similar research has emerged in pathology images. Some studies used meta-learning methods, such as MAML [10, 42], prototypical networks [50, 9], and matching networks [54], for tasks like whole-genome doubling prediction [6] and cancer classification [49, 45]. Limited by the scale of

pre-training data, these methods only achieve limited generalizability. Another group of studies borrowed the idea of fine-tuning V-L models, with related research still in the early stages. For example, CITE [62] applied visual prompt fine-tuning techniques to few-shot learning in pathology images. CLIPath [24] fine-tuned a learnable network layer on top of a frozen foundation model to transfer foundation model knowledge. However, these studies are only applicable to small-sized pathology image patches and cannot directly handle entire WSI. Qu et al. [38] leveraged the powerful generalization capabilities of CLIP, successfully achieving WSI classification tasks using only slide labels. Limited by the weakly supervised slide label, there still exists a significant gap in classification accuracy compared to fully supervised methods. PLIP [16] and CONCH [34] fine-tuned multimodal large models like CLIP [41] and CoCa [56] to perform pathology classification tasks. However, they still rely on large-scale pathology image-text pair datasets for effective performance. Different from previous works, we propose a dual-level few-shot annotation strategy and a dual-tier few-shot learning formulation approach for WSI classification, which balances annotation cost and granularity, achieving excellent classification accuracy close to fully supervised methods.

**Vision-Language Model Adaptation** V-L foundation models such as CLIP [41], ALIGN [21], and Florence [57] have demonstrated remarkable generalization capabilities. How to fine-tune these V-L foundation models for adaptation to downstream tasks is crucial. The popular adaptation strategies can be divided into two groups: prompt tuning and feature adapter. CoOp [63] is a representative method of prompt tuning, which optimizes a set of learnable prompt tokens to enhance the performance of V-L models in downstream tasks. A representative method of feature adapter is CLIP-Adapter [13], which fine-tunes the CLIP model by adding a lightweight residual module after the encoder. Furthermore, Tip-Adapter [61] constructs a key-value cache model to integrate the knowledge from a few-shot training set directly into the CLIP model, effectively speeding up model convergence during fine-tuning. In the field of natural images, many subsequent works based on Tip-Adapter have also made significant contributions to the development of foundation model adaptation. For example, CaFo[60] effectively combines the different prior knowledge of various pre-trained models by cascading multiple foundation models. CO3[46] goes a step further by considering both general and open-world scenarios, designing a text-guided fusion adapter to reduce the impact of noisy labels. Similarly, for open-world few-shot learning, DeIL[47] proposes filtering out less probable categories through inverse probability prediction, significantly improving performance. APE[64] proposes an adaptive prior refinement method that significantly enhances computational efficiency while ensuring high-precision classification performance. Due to the huge size and the lack of pixel-level annotations, these methods cannot effectively solve the classification problem of WSIs. However, existing methods are only applicable to fully annotated data and cannot effectively utilize unannotated patches in pathology images. Our work is inspired by Tip-Adapter, but it differs from Tip-Adapter. The key-value cache model built by Tip-Adapter only allows the key to be learnable. To fully utilize all patches in WSIs, we built a cache model where keys and values are learnable. This effectively facilitates the learning of correct label information for a large number of unlabeled patches, significantly enhancing the performance of CLIP for WSI classification.

## 3 Method

In this chapter, we first explain how to efficiently annotate WSIs, then describe how to formulate the few-shot WSI classification problem under the novel annotation strategy, and finally introduce the classification framework of FAST.

### 3.1 Annotation Strategy and Problem Formulation

For a better understanding, we first present a fully labeled WSI dataset. Given a dataset $X_{train} = \{X_1, X_2, \ldots X_M\}$ consisting of $M$ WSIs, where each WSI $X_i = \{x_{i,1}, x_{i,2}, \ldots, x_{i,U_i}\}$ consisting of $U$ patches. Each patch $x_{i,j}$ is considered an instance, and all patches in $X_i$ form a bag. For $X_i$, we have a bag-level label $Y_i^B$, and $Y_{train}^B = \{Y_1^B, Y_2^B, \ldots Y_M^B\}$. For $x_{i,j}$, we have an instance-level label $y_{i,j}$, and $Y_{train}^I = \{\{y_{1,1}, y_{1,2}, \ldots, y_{1,U_1}\}, \{y_{2,1}, y_{2,2}, \ldots, y_{2,U_2}\}, \ldots, \{y_{M,1}, y_{M,2}, \ldots, y_{M,U_M}\}\}$. Similarly, for the testing set, the data is denoted as $X_{test}$, and the labels consist of $Y_{test}^B$ and $Y_{test}^I$.

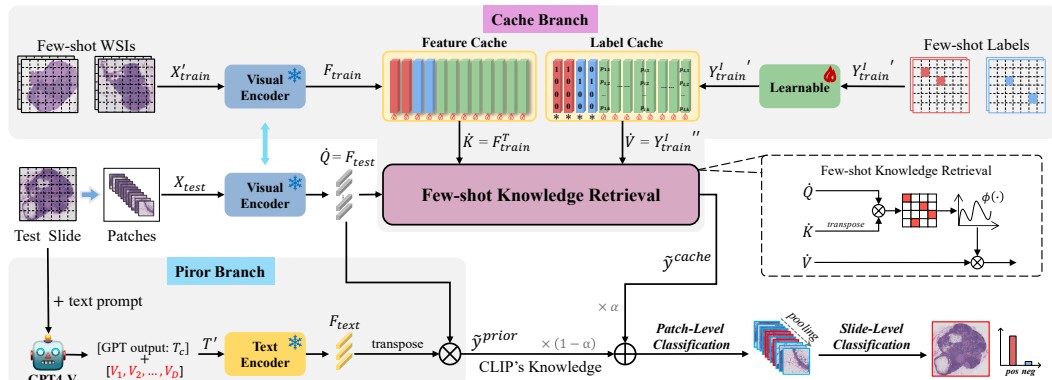

Figure 2: The structure of the FAST classification framework.

As shown in Figure 1(c), the steps of the dual-tier few-shot annotation strategy are as follows: firstly, we collect a small number of WSIs $X_{train}' = \{X_1, X_2, \ldots X_K\}$, where $K$ is much smaller than $M$. Then, for each WSI in $X_{train}'$, only $L$ patches are labeled, resulting in $Y_{train}^{I}{}' = \{\{y_{1,1}, y_{1,2}, \ldots, y_{1,L}\}, \{y_{2,1}, y_{2,2}, \ldots, y_{2,L}\}, \ldots, \{y_{K,1}, y_{K,2}, \ldots, y_{K,L}\}\}$, where $L$ is much smaller than $U$. In this annotation strategy, the size of $X_{train}'$ is much smaller than $X_{train}$, and the number of instance labels $Y_{train}'$ is much smaller than $Y_{train}$.

In conventional few-shot learning, a "N-way K-shot" training set is provided, where N-way represents N categories, and K-shot represents K labeled samples. Under our dual-tier few-shot annotation strategy, the 'shot' includes bag-level few-shot and instance-level few-shot. Therefore, we formulate the few-shot WSI classification task as "N-way K-bshot & L-ishot", where K-bshot represents K bags, and L-ishot represents L labeled instances. During training, only $X_{train}'$ and $Y_{train}'$ are used for training, but during testing, the complete test dataset $X_{test}$ and $Y_{test}$ are used for testing.

## 3.2 Classification Framework

As shown in Figure 2, the classification framework consists of two branches: a learnable image cache branch and a CLIP prior knowledge branch. During training, only a very small number of instances need to be annotated to train the cache model and the learnable prompt tokens. During testing, the prediction results from both the image cache branch and the CLIP prior knowledge branch are integrated to obtain instance-level classification results. Finally, the instance-level classification results are pooled to yield a bag-level classification result.

**1) Image Cache Branch** The cache model consists of a feature cache and a label cache, and its construction method is illustrated in Figure 2. First, all patches from these WSIs are input into the image encoder for feature extraction, and the extracted features are stored in the feature cache.

$$F_{train} = \text{VisualEncoder}\left(X_{train}'\right)$$

The VisualEncoder represents the image encoder within CLIP. Simultaneously, the labels of the annotated instances are transformed into one-hot encoding and stored in the label cache. For the remaining instances without labels, we set their labels as learnable parameters $P_{train}$ and store them in the label cache.

$$Y_{train}^{I} = \text{Cat}\left(Y_{train}^{I}{}', P_{train}\right)$$

$P_{train} = \{\{p_{1,L+1}, \ldots, p_{1,L+K_1}\}, \{p_{2,L+1}, \ldots, p_{2,L+K_2}\}, \ldots, \{p_{i,L+1}, \ldots, p_{i,L+K_i}\}\}$ represents the pseudo-labels of all unannotated instances in $X_{train}'$, where $p$ is a learnable high-dimensional vector. As illustrated in the few-shot knowledge retrieval module in Figure 2, through knowledge retrieval, we can obtain the prediction of the cache branch. Specifically, we employ an attention mechanism to implement the knowledge retrieval module, and treat the features and labels as key-value pairs. The Features $F_{train}$ serve as keys, denoted by $\dot{K}$. The labels $Y_{train}^{I}$ serve as values, denoted by $\dot{V}$. The feature of patch to be retrieved serve as query, denoted by $\dot{Q}$. The retrieval result is $\phi(\dot{Q}\dot{K}^T)\dot{V}$, where $\phi(\cdot) = softmax(\cdot)$.

We train the cache model using $F_{train}$ and $Y_{train}'$. According to the idea of knowledge retrieval, given a query instance $x_{train} \in X_{train}$ and its encoded feature $f_{train} \in F_{train}$, and the prediction

result obtained from the few-shot knowledge retrieval is $\tilde{y}^{cache} = f_{train}F_{train}^{T}Y_{train}^{I}{}''$. Subsequently, the cross-entropy between the predicted values $\tilde{y}_{i,j}^{cache}$ from the cache model and the ground truth $y_{i,j}$ is calculated as the loss function.

$$CacheLoss_{i,j} = \text{CE}\left(\tilde{y}_{i,j}^{cache}, y_{i,j}\right) = \text{CE}\left(f_{i,j}F_{train}^{T}Y_{train}^{I}{}'', y_{i,j}\right)$$

The learnable labels in the cache model can be optimized by minimizing the loss function. Additionally, to further optimize the feature space of CLIP, we also set $F_{train}$ to be learnable as follow.

$$P_{train}^{*} = \min_{P_{train}}\sum_{i,j}CacheLoss_{i,j}, \quad F_{train}^{*} = \min_{F_{train}}\sum_{i,j}CacheLoss_{i,j}$$

The parameters of the VisualEncoder are frozen. It is important to note that in practical training, when there is an excess of unlabeled data, incorporating all the unlabeled instances into the cache model can exceed memory constraints. In such cases, we use K-means clustering algorithm to select representative instances to construct a core set. Only the core set is incorporated into the cache model for training and inference.

**2) CLIP Prior Knowledge Branch** To further leverage the prior knowledge of the vision-language foundation model, we construct a prompt-learnable vision-language instance classifier based on CLIP's text encoder. Since the feature spaces of CLIP's text encoder and image encoder are aligned, image classification can be achieved by calculating the similarity between text and images. However, unlike natural image classification, pathological image classification requires more specialized and targeted prompts. As shown in the prior branch in Figure 2, we input a small number of annotated instance images into GPT-4V, which generates descriptions of the images combining relevant pathological knowledge, forming prompts for detecting each type of WSI. Then, these category-specific prompts are feature-extracted by CLIP's text encoder,

$$f_c^{text} = \text{TextEncoder}\left(T_c\right), \quad T_c = [W]_{c,1}[W]_{c,2}\ldots$$

where $f_c^{text}$ is the text feature for category $c$, $f_c^{text} \in F_{text}$. $T_c$ is the prompt encoding for category $c$, consisting of multiple word vectors $[W]$. Finally, given the CLIP-encoded test image feature $f_{test}$, the classification result under $N$-class text descriptions is,

$$\tilde{y}^{prior} = p\left(y = c \mid f_{test}\right) = \frac{\exp\left(\cos\left(f_c^{text}, f_{test}\right)/\tau\right)}{\sum_j^N \exp\left(\cos\left(f_j^{text}, f_{test}\right)/\tau\right)}$$

where $\tau$ is the temperature coefficient, learned by CLIP during the pre-training phase.

Furthermore, inspired by CoOp, we make the category-specific prompts learnable. Specifically, we add $D$ learnable tokens to the prompts generated for each category, combining them into the final learnable prompts $T'$. The new text feature is as follows,

$$f_c^{text'} = \text{TextEncoder}\left(T_c'\right), \quad T_i' = [W]_{i,1}[W]_{i,2}\ldots[V]_{i,1}[V]_{i,2}\ldots[V]_{i,D}$$

Based on the new text feature and the image feature $f_{i,j}$, the classification result of prior branch is,

$$\tilde{y}_{i,j}^{prior} = p\left(y_{i,j} = c \mid f_{i,j}\right) = \frac{\exp\left(\cos\left(f_c^{text'}, f_{i,j}\right)/\tau\right)}{\sum_j^N \exp\left(\cos\left(f_j^{text'}, f_{i,j}\right)/\tau\right)}$$

Similar to the image cache branch, we also use all annotated instances to train the prior knowledge branch. The optimization function and loss function of the prior branch are as follows,

$$[V] = \min_{[V]}\sum_{i,j}TextLoss_{i,j}, \quad \text{where } TextLoss_{i,j} = \text{CE}\left(\tilde{y}_{i,j}^{prior}, y_{i,j}\right)$$

The parameters of the TextEncoder are also frozen. Finally, we combine the predictions and loss functions from the image cache branch and the prior knowledge branch to obtain the overall model's instance-level classification result and loss function,

$$\tilde{y} = \alpha \cdot \tilde{y}^{cache} + (1 - \alpha) \cdot \tilde{y}^{prior}$$

$$Loss = \sum_{i,j}CacheLoss_{i,j} + \sum_{i,j}TextLoss_{i,j}$$

where $\alpha$ is the fusion weight of the two branches, which can be considered a hyperparameter. We divide the fusion weight $\alpha$ into equal intervals with a step size of 100, then sequentially calculate the classification accuracy for each fusion ratio, and finally select the fusion weight that yields the highest classification accuracy as the fusion weight $\alpha$ for this task. Since the entire network has few parameters and fast inference speed, this parameter can be quickly optimized through grid search to obtain the optimal value.

## 4    Experiments

### 4.1    Datasets and Few-Shot Scenario Simulation

We evaluated our method on the public WSI datasets CAMELYON16 [4] and TCGA-RENAL [17]. CAMELYON16 is a binary dataset used to detect whether it contains breast cancer metastasis. TCGA-RENAL is a multi-class dataset for classifying renal cancer subtypes, covering clear cell renal cell carcinoma (ccRCC), papillary renal cell carcinoma (pRCC), and chromophobe renal cell carcinoma (chRCC). For more details about the datasets, please refer to the supplementary material A.1. For few-shot scenario simulation, we first simulate the collection of few-shot WSI data in clinical settings by sampling 1, 2, 4, 8, and 16 WSIs for each category. Then, we simulate the process of expert annotation of few-shot instances within sampled bags, initially selecting 10% instances as a core set by K-means clustering [2], then randomly sampling 16 instances per category from the core set as labeled instances, with others remaining unlabeled. Through this simulation, for CAMELYON16 and TCGA-RENAL datasets, we obtained few-shot training datasets with 1, 2, 4, 8, and 16 bag shots and 16 instance shots. Considering the randomness in few-shot learning, we conducted five random samplings and model trainings for all scenarios and reported the mean and variance of classification results.

### 4.2    Comparing Methods and Evaluation Metrics

First, we compared FAST with zero-shot learning method CLIP [36] and fully supervised method using all instance-level labels [35], which can respectively be seen as the lower and upper bounds of deep learning method performance in few-shot WSI classification. Subsequently, due to lack of effective few-shot WSI learning methods, we compared the latest few-shot learning methods Tip-Adapter and Tip-Adapter-F [61] in natural images. Bag-level weakly supervised multi-instance learning methods, such as R2T[52], generally perform worse than instance-level fully supervised methods due to the lack of fine-grained labels. Therefore, this paper does not directly compare with multi-instance learning methods. For implementation details of our method, please refer to the supplementary material A.2. To provide a comprehensive evaluation of these methods, we reported instance-level Area Under Curve and bag-level Area Under Curve (AUC) [5]. Since the TCGA-RENAL dataset is a multi-class classification task, we reported instance-level and bag-level AUCs separately for each category.

### 4.3    Experimental Results

**CAMELYON16**    The few-shot WSI classification results on the CAMELYON16 dataset are shown in Table 1. **Firstly**, in the zero-shot setting, the instance-level AUC and bag-level AUC of zero-shot CLIP are very low, almost unable to handle the classification task. In contrast, under extreme few-shot setting with 1 bag shot and 16 instance shots, FAST achieves 0.84 instance-level AUC, which shows a significant improvement over zero-shot CLIP. **Secondly**, as the training samples increase, the performance of FAST shows a significant improvement trend. Meanwhile, our proposed method FAST consistently outperforms the comparison methods Tip-Adapter and Tip-Adapter-F across different numbers of samples. Although Tip-Adapter and Tip-Adapter-F have achieved significant success in natural images, their performance is poor on few-shot WSI classification task, especially with bag-level AUC generally below 0.6. We think there are two main reasons for this: (1) there are domain differences between pathology images and natural images, making it difficult for the text branch of CLIP to accurately classify instances and even more difficult to classify bags. (2) The limited WSIs make it more difficult to ensure the diversity of instances sampled from these WSIs, resulting in poor generalization of these methods. In contrast, while FAST also utilizes a small number of bags, it fully leverages the labeled and unlabeled instances within these bags through a learnable label cache, enabling it to learn more comprehensive instance representations. **Finally**,

Table 1: Results on CAMELYON16 dataset

| Bag Shot | Instance Shot | Methods | Instance-level AUC | Bag-level AUC |
|---|---|---|---|---|
| 0 | 0 | Zero-shot CLIP | 0.6711 | 0.5409 |
| 0 | 0 | Zero-shot PLIP | 0.6004 | 0.5434 |
| 0 | 0 | Zero-shot CONCH | 0.8929 | 0.7113 |
| 1 | 16 | **FAST** | **0.8400±0.0335** | **0.6933±0.0846** |
| | | Tip-Adapter-F | 0.7162±0.0435 | 0.5653±0.0604 |
| | | Tip-Adapter | 0.6275±0.0777 | 0.498±0.0187 |
| 2 | 16 | **FAST** | **0.8584±0.0380** | **0.7595±0.0391** |
| | | Tip-Adapter-F | 0.7200±0.0595 | 0.5748±0.0537 |
| | | Tip-Adapter | 0.6198±0.0823 | 0.5141±0.0156 |
| 4 | 16 | **FAST** | **0.8864±0.0563** | **0.7359±0.0853** |
| | | Tip-Adapter-F | 0.6990±0.0890 | 0.5731±0.0401 |
| | | Tip-Adapter | 0.5601±0.0772 | 0.5321±0.0152 |
| 8 | 16 | **FAST** | **0.9060±0.0074** | **0.7742±0.0249** |
| | | Tip-Adapter-F | 0.7392±0.0180 | 0.6045±0.0044 |
| | | Tip-Adapter | 0.6782±0.0166 | 0.4880±0.0097 |
| 16 | 16 | **FAST** | **0.9151±0.0200** | **0.8197±0.0474** |
| | | Tip-Adapter-F | 0.7227±0.0098 | 0.5965±0.0243 |
| | | Tip-Adapter | 0.6835±0.0135 | 0.4913±0.0164 |
| All | All | Fully Supervised | 0.9532 | 0.8555 |

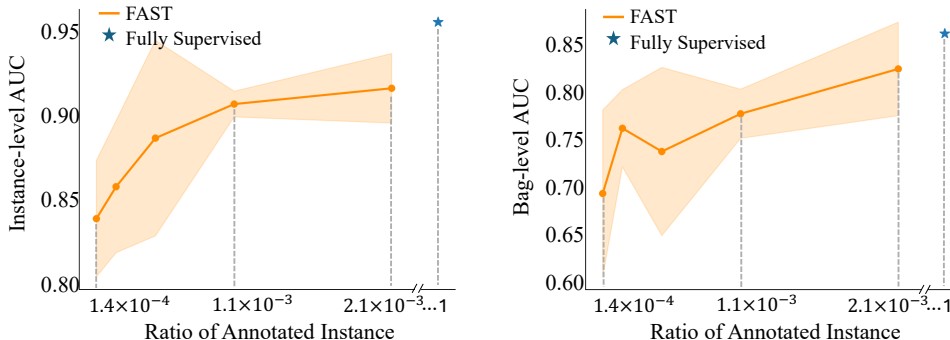

Figure 3: Results of FAST on CAMELYON16 dataset under different annotation ratio.

in the setting where only 16 bag shots and 16 instance shots are available, our proposed FAST method achieves 0.9151 instance-level and 0.8197 bag-level AUC, which is close to the performance of the fully supervised method using all instance annotations. Importantly, the annotation cost of FAST is only 0.22% of that of the fully supervised method, as detailed in Section 4.4. **In summary**, FAST achieves accuracy close to the fully supervised method with extremely low data collection and annotation costs. This significantly enhances the efficiency of establishing deep learning models in clinical settings and opens up possibilities for widespread clinical adoption of WSI classification algorithms based on deep learning.

**TCGA-RENAL**  In the few-shot experiments on the TCGA-RENAL dataset, the classification results are shown in Table 2. Compared to the binary classification task on the CAMELYON16 dataset, the three-class classification task is more challenging. Therefore, the zero-shot CLIP method only achieves around 0.5 instance-level AUC, almost unable to handle the classification task. In the setting with very few shots, such as one or two bags, the results of all methods are relatively poor, with instance-level AUC and bag-level AUC both less than 0.7000. However, our proposed FAST still achieves SOTA performance on most metrics. As the bag shot reaches 4 or more, the classification results begin to significantly improve, and FAST also outperforms other methods on all metrics. Especially in 16 bag shots, FAST significantly outperforms the comparison methods. The average bag-level AUC for chRCC reaches 0.9234, differing by only 0.0033 from the fully supervised method. Similarly, the bag-level AUC for ccRCC and pRCC also reach 0.9254 and 0.9216 respectively, differing by only 0.0218 and 0.036 from the fully supervised method. This experimental result further demonstrates that, in complex multi-class classification tasks, FAST can approach the fully supervised method with very low annotation costs, achieving the state-of-the-art performance in few-shot learning.

Table 2: Results on TCGA-RENAL dataset

| Bag Shot | Instance Shot | Methods | Instance-level AUC | | | Bag-level AUC | | | |
|---|---|---|---|---|---|---|---|---|---|
| | | | ccRCC | pRCC | chRCC | ccRCC | pRCC | chRCC | mean |
| 0 | 0 | Zero-shot CLIP | 0.5475 | 0.5521 | 0.3640 | 0.4959 | 0.5192 | 0.4811 | 0.4987 |
| 0 | 0 | Zero-shot PLIP | 0.5396 | 0.6006 | 0.4774 | 0.6036 | 0.6610 | 0.4905 | 0.5850 |
| 0 | 0 | Zero-shot CONCH | 0.8127 | 0.9122 | 0.9131 | 0.9039 | 0.8936 | 0.9449 | 0.9141 |
| 1 | 16 | **FAST** | **0.5935±0.0488** | **0.6853±0.0443** | **0.6548±0.0760** | **0.6067±0.0833** | **0.6921±0.0416** | 0.6488±0.0984 | **0.6492** |
| | | Tip-Adapter-F | 0.5710±0.0387 | 0.6533±0.0566 | 0.6230±0.0967 | 0.5981±0.0393 | 0.6523±0.0684 | **0.6948±0.1113** | 0.6484 |
| | | Tip-Adapter | 0.5874±0.0503 | 0.6308±0.0655 | 0.6017±0.0734 | 0.5951±0.0602 | 0.6515±0.0668 | 0.6624±0.1099 | 0.6363 |
| 2 | 16 | **FAST** | **0.6245±0.0733** | **0.6998±0.0229** | **0.6987±0.0582** | **0.6745±0.0900** | 0.7327±0.0279 | 0.7329±0.0483 | **0.7133** |
| | | Tip-Adapter-F | 0.6084±0.0609 | 0.6820±0.0405 | 0.6985±0.0481 | 0.6359±0.1143 | **0.7391±0.0331** | **0.7481±0.0794** | 0.7077 |
| | | Tip-Adapter | 0.6068±0.0524 | 0.6413±0.0554 | 0.6238±0.0522 | 0.6545±0.0931 | 0.6682±0.1034 | 0.7405±0.0763 | 0.6877 |
| 4 | 16 | **FAST** | **0.7107±0.1056** | **0.7547±0.0544** | **0.7652±0.0645** | **0.7684±0.1681** | **0.8260±0.0816** | **0.8143±0.1080** | **0.8029** |
| | | Tip-Adapter-F | 0.6587±0.0858 | 0.7266±0.0756 | 0.7488±0.0555 | 0.7220±0.1091 | 0.7621±0.0985 | 0.7132±0.1631 | 0.7324 |
| | | Tip-Adapter | 0.6361±0.0737 | 0.6797±0.0820 | 0.6835±0.0805 | 0.6671±0.1384 | 0.7274±0.0391 | 0.7856±0.0866 | 0.7267 |
| 8 | 16 | **FAST** | **0.7940±0.0522** | **0.8228±0.0275** | **0.8398±0.0194** | **0.8955±0.0453** | **0.8978±0.0478** | **0.8964±0.0485** | **0.8966** |
| | | Tip-Adapter-F | 0.7249±0.0529 | 0.7832±0.0255 | 0.7918±0.0239 | 0.7854±0.1382 | 0.8239±0.0595 | 0.7879±0.0595 | 0.7991 |
| | | Tip-Adapter | 0.6839±0.0567 | 0.7552±0.0275 | 0.7623±0.0380 | 0.7735±0.0793 | 0.7787±0.0526 | 0.7908±0.0735 | 0.7810 |
| 16 | 16 | **FAST** | **0.8252±0.0428** | **0.8420±0.0126** | **0.8609±0.0165** | **0.9254±0.0206** | **0.9216±0.0233** | **0.9234±0.0184** | **0.9235** |
| | | Tip-Adapter-F | 0.7409±0.0736 | 0.8026±0.0151 | 0.8030±0.0268 | 0.8612±0.0726 | 0.8235±0.0547 | 0.8531±0.0591 | 0.8459 |
| | | Tip-Adapter | 0.6911±0.0484 | 0.7886±0.0230 | 0.7851±0.0191 | 0.7624±0.0931 | 0.7593±0.0400 | 0.8877±0.0621 | 0.8031 |
| All | All | Fully Supervised | 0.8917 | 0.9055 | 0.8977 | 0.9472 | 0.9576 | 0.9267 | 0.9438 |

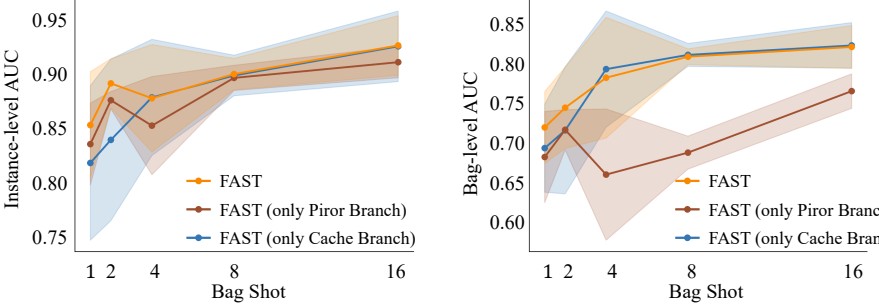

Figure 4: Comparison of cache branch and prior branch in FAST.

## 4.4 Annotation Efficiency

To illustrate the annotation efficiency of FAST, we compared the classification results of our method and the fully supervised method under different annotation ratios. For the CAMELYON16 dataset, the results are presented in Figure 3. It can be observed that the classification AUC of FAST rapidly increase with the growth of bag shots. When the bag shot reaches 16, the annotation only accounts for 0.22%. The average bag-level AUC reaches 96.32% of the fully supervised method. This result fully demonstrates the advantage of FAST in annotation efficiency. For the results on the TCAG-RENAL dataset, please refer to the supplementary material A.3.

## 4.5 Ablation Studies

*To analyze the importance of each component in FAST*, we conducted a serise of experiments under the conditions of 16 bag shots and 16 instance shots on the CAMELYON16 dataset, including whether to use a cache branch, whether to set the feature cache as learnable, whether to use a prior branch, and whether to utilize unlabeled instances to construct a learnable label cache. The results of the ablation experiments are shown in Table 3. The first three rows of the table correspond to zero-shot CLIP, Tip-Adapter, and Tip-Adapter-F, respectively. The next three rows correspond to FAST using only the prior branch, FAST using only the cache branch, and the complete FAST. **Firstly**, the first three rows of the table show that the instance-level AUC gradually increases from 0.6711 to 0.7277 for the three methods. This indicates that the cache branch can leverage a small amount of supervised information, and making the cache feature learnable can further optimize the feature space distribution of the cache model. However, even the best model's instance-level AUC is only 0.7227, and the pooled bag-level AUCs are all less than 0.6. **Next**, from the last three rows of the table, even with only the prior branch, using our proposed prompt-based method, the instance-level and bag-level classification AUCs reach 0.8739 and 0.7931, respectively, which outperformed the results of Tip-Adapter-F. The instance-level and bag-level AUCs of only using the cache branch can reach 0.9165 and 0.8183, respectively, demonstrating that the cache branch is crucial for FAST. By further comparing the third and fifth rows of the table, we can infer the primary reason why FAST's cache model surpasses that of

Table 3: Ablation study of FAST on CAMELYON16 dataset

| Cache Branch | Learnable Feature Cache | Prior Branch | Learnable Label Cache | Instance-level AUC | Bag-level AUC |
|---|---|---|---|---|---|
| | | | | 0.6711 | 0.5409 |
| ✓ | | | | 0.6835±0.0135 | 0.4913±0.0164 |
| ✓ | ✓ | | | 0.7227±0.0098 | 0.5965±0.0243 |
| | | ✓ | | 0.8739±0.0161 | 0.7931±0.0247 |
| ✓ | ✓ | | ✓ | **0.9165±0.0213** | 0.8183±0.0560 |
| ✓ | ✓ | ✓ | ✓ | 0.9151±0.0200 | **0.8197±0.0474** |

Tip-Adapter-F in natural images. Specifically, FAST utilizes a large number of unannotated instances in a small number of bags through the learnable label cache, thereby significantly improving the capacity and generalization ability of the cache model. Finally, comparing the last two rows of the table, we find that when both cache and prior branches are utilized, FAST does not show significant improvement over using only the cache branch. This is because the number of samples is sufficient, and the prior branch primarily plays a role when there are fewer samples.

*To verify the role of two branchs*, we compared the prior branch and cache branch of FAST under different bag shots. The results are shown in Figure 4. When there are only 1 or 2 bags, the instance classification results of the prior branch are significantly higher than those of the cache branch. The instance and bag classification results combined with both the cache and prior branches also surpass those of using each branch separately, indicating that the prior branch performs better in extreme samples, and the information learned by the prior branch and the cache branch is complementary. As the number of bags increases to 4 or more, the results of the cache branch gradually surpass those of the prior branch. Therefore, in extreme few-shot scenarios, FAST is dominated by the prior branch, but as the sample size gradually increases, FAST is dominated by the image branch. This experimental analysis can provide effective guidance for the practical application of FAST. Additionally, we provide further analysis experiments on the number of annotated instance and the number of core sets in the supplementary material A.3.

## 5 Conclusion

In this paper, we propose a dual-tier few-shot learning paradigm FAST for WSI classification, which achieves low-cost WSI annotation, high-accuracy WSI classification, and adaptation to various WSI classification tasks. To achieve these goals, we introduce two key technologies in FAST, including a dual-level few-shot annotation strategy and a dual-branch classification framework. The dual-level few-shot annotation strategy effectively alleviates the problem of fine-grained annotation difficulty in WSI by annotating only a small number of patches in a limited number of WSIs. Experimental results demonstrate that the annotation cost of our method is only 0.22% of patch-level full annotation. In the dual-branch classification framework, we construct a cache branch where both features and labels are learnable, fully exploiting the partially annotated data obtained from the dual-level few-shot annotation strategy. Furthermore, combining a vision-language foundation model and prompt tuning technology, we build a prior knowledge branch to assist the cache branch in improving classification performance. Through extensive experiments, we demonstrate that FAST can achieve state-of-the-art performance on both binary and multi-class classification tasks. However, our method still has certain limitations. For bag-level classification, we simply use pooled instance classification results, ignoring the relationships between instances. In the future, we will further explore how to use instance-level classification results to obtain a better bag-level classification result.

## Acknowledgments and Disclosure of Funding

This work was supported by the National Key R&D Program of China under Grant No. 2022ZD0116800, and the National Natural Science Foundation of China under Grant 62471149 and Taishan Scholars Program under Grant TSQN202211214.

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

# A    Supplemental Material

## A.1    Datasets

**CAMELYON16**    CAMELYON16 [4] contains 400 WSIs of lymph nodes, used to detect the presence of metastatic breast cancer. WSIs containing metastases are labeled positive, while others are negative, with pixel-level annotations of the metastatic areas. We first cropped these WSIs at 10x magnification into 512x512 image patches, then removed image patches with entropy less than 15 as background, and marked patches with more than 30% cancer area as positive, resulting in 186,604 image patches, of which 8,117 were marked as positive (4.3%).

**TCGA-RENAL**    This dataset comes from The Cancer Genome Atlas (TCGA) project [17], covering high-resolution WSIs of the three main subtypes of renal cell carcinoma (RCC): clear cell renal cell carcinoma (ccRCC), papillary renal cell carcinoma (pRCC), and chromophobe renal cell carcinoma (chRCC). ccRCC is the most common subtype, characterized by clear cells containing abundant lipids and glycogen. pRCC follows, characterized by papillary structures formed on the surface of tumor cells. chRCC is relatively rare, with larger cells and granular cytoplasm. All WSIs are rigorously reviewed by professional pathologists to ensure the representativeness and quality of the data. To fully verify the effectiveness of our proposed method, we collected 910 WSIs from TCGA, and then organized experts to delineate all WSIs at the pixel level, including delineation of cancerous and non-cancerous areas. Subsequently, through preprocessing processes such as WSI segmentation and background removal similar to CAMELYON16, we obtained instance-level labels for the complete training and testing sets. Finally, all WSIs were divided into training and testing sets in a ratio of 70% and 30%, respectively.

## A.2    Implementation Details

FAST employs the image encoder and text encoder from the pre-trained CLIP-RN50 [41]. The cache capacity is typically determined by the number of annotated and unannotated instances in the training set after each sampling. For bag shot greater than 4, we used a core set selection strategy to prevent excessive caching. For the CAMELYON16 and TCGA-RENAL datasets, we set the number of core sets to 1000 and 2000, respectively. We set the number of learnable tokens to 10. During training, we utilized the Adam optimizer with learning rates set as follows: 0.001 for the feature cache, 0.01 for the label cache, and 0.001 for the tokens in the prior branch. We fine-tune our model with batch size of 4096 for 20,000 epochs. All models are trained and tested on an RTX 3090 GPU with 24GB memory.

For Tip-Adapter, We conducted comparative experiments according to the settings of the optimal model in the original Tip-Adapter paper. For aspects that cannot be adapted to the few-shot WSI classification task, we used the following approach. We designed a set of text prompts specifically for pathology images, which has been proven superior in the CONCH comparison experiments we conducted. We used all annotated patches to build the cache model.

## A.3    Additional experiments

**(1) Annotation Efficiency**    To provide a more intuitive illustration of the annotation efficiency of our method, we first calculated the annotation ratio under different bag shots (1, 2, 4, 8, 16), with each bag containing 16 annotated instances. Then, we visualized the classification results of FAST under different annotation ratios and compared them with the fully supervised method. The results for TCGA-RENAL dataset are presented in Figure 5. It can be observed that the classification AUC of FAST rapidly increase with the growth of bag shots. When the bag shot reaches 8, FAST achieves results comparable to fully supervised methods. At this point, the annotation ratio of FAST on the TCGA-RENAL dataset is only 0.0067% of the total instance annotation. When the bag shot reaches 16, the annotation only accounts for 0.013% of all instances. Even under such extreme minimal annotation, FAST can still achieve results close to fully supervised methods, especially in bag classification results. For chRCC, pRCC, and ccRCC, FAST's average bag-level AUC reaches 99.64%, 96.24%, and 97.70% respectively compared to the fully supervised method. This result fully demonstrates the advantage of FAST in annotation efficiency.

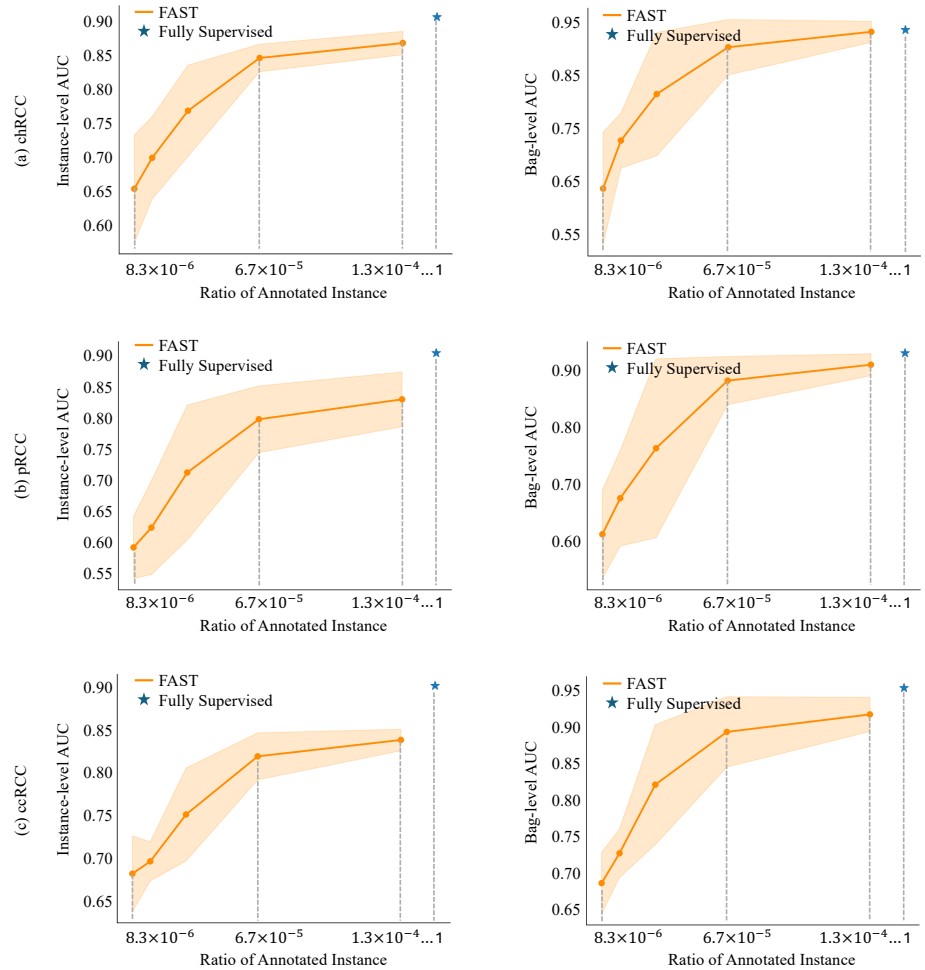

Figure 5: Results of FAST on TCGA-RENAL dataset under different annotation ratio.

**(2) The Number of Annotated Instance**   The above experiments indicate that achieving good results only require annotating 16 instances per bag. To further investigate the influence of the number of annotated instances per bag, we conducted experiments on the CAMELYON dataset with the number of annotated instances per bag set to 4, 16, and 64, respectively. The results are shown in Figure 6. It can be observed that when the number of bags is small and there are only 1, 2, or 4 bags per class, the instance-level and bag-level classification AUCs are significantly higher when 64 instances are annotated per bag compared to only annotating 4 or 16 instances. However, as the number of bags increases to 8 and 16, the AUC of FAST gradually converge. Regardless of whether each bag is annotated with 4, 16, or 64 instances, FAST achieves similar results in instance-level AUC and bag-level AUC. These experimental results indicate that when the number of bags is extremely small, increasing the number of annotated instances can effectively improve the performance of FAST. However, when the number of bags increases to a certain level, even if only a small number of instances are annotated in each bag, FAST can achieve results similar to annotating a large number of instances. This indicates that FAST's ability to achieve higher accuracy than other methods primarily relies on its learning from unlabeled instances. It also suggests that in practical applications of FAST, if there are a large number of bags, the requirement for labeling instances can be appropriately reduced.

**(3) Core Set Size**   Few-shot learning of WSI classification differs from conventional few-shot learning. Even with only several WSIs, they may produce tens of thousands or even hundreds of thousands of patches. Therefore, to avoid optimization difficulties caused by excessively large caches,

Table 4: Results of FAST on the CAMELYON16 dataset under different core set sizes

| Bag Shot | Instance Shot | Core Set Size | Instance-level AUC | Bag-level AUC |
|---|---|---|---|---|
| 4 | 16 | 100 | 0.8845±0.0525 | 0.7295±0.0743 |
| | | 500 | 0.8770±0.0643 | 0.7210±0.1157 |
| | | 1000 | 0.8860±0.0546 | 0.7554±0.0766 |
| | | 2000 | 0.8985±0.0279 | **0.7595±0.0478** |
| | | 5000 | **0.9027±0.0342** | 0.7499±0.0641 |
| 8 | 16 | 100 | 0.8827±0.0303 | 0.7179±0.0712 |
| | | 500 | 0.8903±0.0162 | 0.7549±0.0445 |
| | | 1000 | 0.8957±0.0132 | 0.7773±0.0319 |
| | | 2000 | 0.9075±0.0122 | 0.7848±0.0392 |
| | | 5000 | **0.9101±0.0165** | **0.8025±0.0237** |
| 16 | 16 | 100 | 0.9008±0.0320 | 0.7353±0.0624 |
| | | 500 | 0.9046±0.0301 | 0.7924±0.0484 |
| | | 1000 | **0.9124±0.0281** | 0.8078±0.0626 |
| | | 2000 | 0.9181±0.0173 | **0.8240±0.0572** |
| | | 5000 | 0.9096±0.0211 | 0.8082±0.0544 |

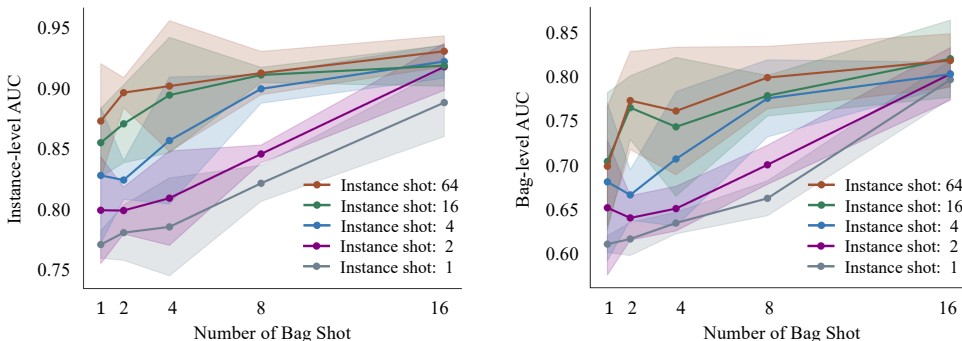

Figure 6: Results of FAST on CAMELYON16 dataset under different instance shots.

FAST employs a K-means core set selection strategy to control the size of the learnable cache. To analyze the performance of FAST under different core set sizes, we conducted experiments on the CAMELYON16 dataset with 4, 8, and 16 bags and 16 instances per bag. The results are shown in Table 4. When the core set size is only 100 or 500, FAST's bag-level AUC is significantly lower, indicating that the core set size at this time is insufficient to cover all representative samples, resulting in a loss of a large amount of information. However, when the core set size reaches 1000 or above, FAST's instance-level AUC and bag-level AUC achieve good results, indicating that FAST has good robustness to core set size. However, when the core set size is 5000 and the number of bags is 16, the results decrease instead, indicating that an overly large learnable cache is difficult to optimize. Therefore, we set the core set size to 1000 or 2000 in experiments where the number of bags is greater than 4.

