# OpenReview forum: "FAST: A Dual-tier Few-Shot Learning Paradigm for Whole Slide Image Classification"
_NeurIPS.cc/2024/Conference — NeurIPS 2024 poster_

### Official Review · Reviewer_x5Ho · 2024-06-17

**Soundness:** 2
**Presentation:** 1
**Contribution:** 2
**Rating:** 5
**Confidence:** 5

**Summary:**

This paper proposes a few-shot learning approach for WSI (Whole-Slide Image) classification. This approach, built upon Tip-Adapter, leverages a cache branch to memorize the knowledge from few-shot instances and then retrieve label information from the cached knowledge. In addition, a prior branch, which utilizes the knowledge from CLIP and GPT4-V, is built to boost the predictive performance. The experiments on two WSI datasets show the superiority of the proposed method over the original Tip-Adapter and Tip-Adapter-F.

**Strengths:**

- Originality. This paper explores the few-shot setting in the context of WSI classification. It is under-studied in the field of computational pathology.
- Significance. This work shows that the proposed approach could obtain a performance near that of fully supervised learning, only with a few labeled instances. Overall, it is an interesting work worthy of investigation in computational pathology, given that labeling WSIs at pixel level is extremely time-consuming and labor-expensive.
- Quality. This work presents a good experimental design. Its experiments are conducted from different angles to verify the effectiveness of the proposed algorithms.

**Weaknesses:**

In summary, my main concerns lie in (1) writing quality, (2) limited technical contribution, and (3) missing experimental comparisons with important vision-language-based models in computational pathology. Based on these critical weaknesses, it may be hard to recommend accepting this paper to NeurlPS. The details are given below:

- This paper is overall rough and sub-par in writing, requiring substantial improvements in clarity. Some obvious flaws are as follows: i) slice-label in line 102, ii) a efficient annotation strategy in line 111, iii) undefined V in line 160, iv) some undefined notations in Section 3.2, and v) some citation errors such as [20]. The authors are encouraged to check these errors and improve their presentation for better academic communication.
- The technical novelty is quite limited. Most key designs of the proposed approach have been proposed in Tip-Adapter (Zhang et al., ECCV 2022). Compared to Tip-Adapter, the proposed approach does not present valuable or substantial technical contributions, since its two crucial components, few-shot knowledge retrieval in the cache branch and the prior branch, seem borrowed from Tip-Adapter.
- The authors claim that their work differs significantly from Tip-Adapter (line 142) because the key-value cache model built by Tip-Adapter only allows the key to be learnable; in contrast, their approach allows both the key and the value to be learnable. The authors are encouraged to rephrase this sentence, as i) also allowing the value to be learnable should not be called a significant modification from my humble understanding; ii) Tip-Adapter actually has proposed to use a learnable value in the key-value cache model but it leads to collapse during training. Moreover, given that the learnable value leads to collapsed training in Tip-Adapter, could the authors explain to readers why their methods can avoid collapsed training?
- Some important vision-language-based models in computational pathology are not cited and compared to the proposed methods, such as **PLIP** (Huang et al., Nature Medicine, 2023) and **CONCH** (Lu et al., Nature Medicine, March 2024). These models show exciting zero-shot performance in WSI classification combined with MI-Zero. The proposed approach should at least show better performance than their zero-shot performance. It could be crucial for justifying the value and significance of this work.

Minor issues:
- It is suggested to rewrite Section 3.2 to make sure that all notations (those in texts and figures) and the implementation of components are clear and well-explained.
- The original Tip-Adapter is proposed for traditional single-instance settings. Its implementation for few-shot WSI classification is not clear.

**Questions:**

- Few-shot instances are randomly selected from the core set. Since only a few instances, e.g. 16, are selected from a very large instance pool and pathological patches (instances) often present heterogeneity, the final selected instance set could have a large variety and thus lead to unstable performances, calling into question the usability of the proposed method. This could also be observed from Fig 3 in the paper. So, the authors are encouraged to analyze and discuss the impact of randomly selected few-shot instances.
- The original TCGA-RCC is not annotated at the pixel level. This work does a good job in terms of annotating the fine-grained region-of-interest of WSIs. However, if the annotation is not made public, this work would be difficult to follow and be very limited in research impact. The authors are encouraged to make their annotated dataset public.

I would like to raise my score if the authors could resolve my concerns & questions above.

---------------------------------------------After Rebuttal-------------------------------------------------

My main concerns have been addressed. I am happy to increase my score.

**Limitations:**

There is no explicit limitation that should be included in the paper.

---

> ### Author Rebuttal · Authors · 2024-08-05
>
> $\textbf{Q1:}$ This paper is overall rough and sub-par in writing, requiring substantial improvements in clarity.
>
> $\textbf{R1: }$ Thanks very much for pointing out the problem. We have revised the above errors as follows: “slice-label” has been corrected to “slide-label,”, “a efficient annotation strategy” has been corrected to “an efficient annotation strategy,”, the symbol “V” has been corrected to “L,” and a “workshops” flag has been added to the citation information for reference [20]. We will conduct sentence-by-sentence proofreading of the paper before the official publication and will also have professionals polish the language.
>
> $\textbf{Q2 and Q3:}$ The technical novelty is quite limited. The authors are encouraged to rephrase this sentence. Moreover, could the authors explain to readers why their methods can avoid collapsed training?
>
> $\textbf{R2 and R3:}$ Thanks for your great suggestion on improving the quality of our manuscript. I believe that the second and third comments in the Weaknesses section are closely related, so we have combined them in our response. I consider the proposed method in this paper to be innovative for the following reasons.
>
> 1. New Paradigm: For the problem of few-shot classification in pathology images, we propose a novel Dual-tier Few-Shot Learning Paradigm, which not only improves classification accuracy but also reduces annotation costs.
>
> 2. Annotation Strategies and Classification Framework: We have introduced a dual-level WSI annotation strategy and a dual-branch classification framework. These two components must work together to achieve excellent WSI classification performance.
>
> 3. Addressing Training Issues: As noted by the reviewer, using learnable keys and values in Tip-Adapter can lead to training instability. Therefore, directly applying a learnable Tip-Adapter to WSI classification is not feasible. Our method ensures that the cache model does not suffer from instability due to its overall design, including the classification paradigm, annotation strategy, and classification framework. First, we obtain precise labels for some patches through the new annotation strategy. Second, in the cache model we construct, the labels of annotated patches are not optimized; only the labels of unannotated patches are optimized. The labels of annotated patches provide correct guidance for learning the labels of other patches, ensuring that the proposed cache model remains stable.
>
> We accept the reviewer’s suggestion “The authors are encouraged to rephrase this sentence.” To clearly state our contributions, we revised “To fully utilize all patches in WSIs, we build a cache model where key and value are learnable, enabling the CLIP to adapt well to the WSI classification task” to “To fully utilize all patches in WSIs, we built a cache model where keys and values are learnable. This effectively facilitates the learning of correct label information for a large number of unlabeled patches, significantly enhancing the performance of CLIP for WSI classification.” At the same time, we rephrased the extent of the differences between our method and Tip-Adapter, revising “Our work is inspired by Tip-Adapter, but it differs significantly from Tip-Adapter.” to “Our work is inspired by Tip-Adapter, but it differs from Tip-Adapter.”
>
> $\textbf{Q4:}$Some important vision-language-based models in computational pathology are not cited and compared to the proposed methods, such as PLIP and CONCH.
>
> $\textbf{R4:}$ Thanks for your great suggestion on improving the quality of our manuscript. We have added comparison experiments with PLIP and CONCH. We used the same experimental setting as Zero-shot CLIP, then employed the PLIP image encoder and text encoder to obtain Zero-shot PLIP, and used the CONCH image encoder and text encoder to obtain Zero-shot CONCH. The experimental results are shown in the following two tables: Table A1 presents the results of the comparison methods with FAST on the CAMELYON16 dataset, and Table A2 shows the results on the TCGA-RENAL dataset.
>
> First, from Table A2, we can see that the bag classification AUC of Zero-shot CONCH that we reproduced on the TCGA-RENAL dataset (referred to as TCGA RCC dataset in CONCH) is 91.94%, which significantly exceeds the 90.2% reported in the CONCH original paper. This indicates that our constructed prompt is very effective and that our Zero-shot CLIP serves as an excellent baseline model. This result also demonstrates that our comparative results across multiple datasets are very valid. From Table A1, it can be observed that our method outperforms PLIP and CONCH in all metrics on the CAMELYON16 dataset. From Table A2, it is evident that our method surpasses PLIP in all metrics on the TCGA-RENAL dataset. Compared to CONCH, FAST significantly outperforms CONCH in the average bag-level classification AUC. It is worth noting that our method uses a maximum of only 16 WSIs, while CONCH relies on 1.17 million pairs of pathology images based on CoCa, resulting in a significantly high training cost.
>
> Last but not least, our method is orthogonal to studies such as PLIP and CONCH, and they do not conflict with each other. For example, in the “Few-shot classification with task-specific supervised learning” section of CONCH, it is stated: “However, it may still be desirable to specialize the model with labeled training examples to maximize performance for a given task, ideally using as few labels as possible. In this section, we investigate the label efficiency when using the pretrained representation of the image encoder backbone of the visual-language foundation models for task-specific supervised classification.”. This indicates that PLIP, CONCH, and similar methods will also need to be combined with few-shot learning methods like FAST proposed in this paper in the future. Therefore, our method can be effectively combined with methods such as PLIP and CONCH to further enhance WSI classification performance.

---

> ### Author Response · Authors · 2024-08-07
> **Part2: Rebuttal by Authors**
>
> $\textbf{Q5:}$ It is suggested to rewrite Section 3.2 to make sure that all notations (those in texts and figures) and the implementation of components are clear and well-explained.
>
> $\textbf{R5:}$ Thanks for your great suggestion on improving the quality of our manuscript. Firstly, we have revised the corresponding errors according to the suggestions in $\textbf{Q1}$. Secondly, we have corrected line 180 from “$\tilde{y}^{\text{cache}} = f_{\text{train}} F_{\text{train}}^T Y_{\text{train}}^I $” to “$\tilde{y}^{\text{cache}} = f_{\text{train}} F_{\text{train}}^T {Y_{\text{train}}^I}^ {{\prime}{\prime}}$”.  We have also added the definition of $P_{\text{train}}$ as follows. $P_{\text{train}} = [\{ p_{(1, L+1)}, p_{(1, L+2)}, \ldots, p_{(1, K_1)}\}, \{ p_{(2, L+1)}, p_{(2, L+2)}, \ldots, p_{(2, K_2)} \}, \ldots, \{ p_{(i, L+1)}, p_{(i, L+2)}, \ldots, p_{(i, K_i)} \} ]$ represents the pseudo-labels of all unannotated instances in ${{X}_{\text{train}}}^{\prime}$, where  $p$  is a learnable high-dimensional vector.
>
>
> $\textbf{Q6:}$ The original Tip-Adapter is proposed for traditional single-instance settings. Its implementation for few-shot WSI classification is not clear.
>
> $\textbf{R6:}$ We apologize for any inconvenience brought to you. We will add the implementation details of the Tip-adapter for few-shot WSI classification tasks to the supplementary materials. Additionally, we will open-source the code to facilitate further research by other researchers. The implementation details are as follows. We conducted experiments according to the settings of the optimal model in the original Tip-Adapter paper. For aspects that cannot be adapted to the few-shot WSI classification task, we used the following approach.
> 1. We designed a set of text prompts specifically for pathology images, which has been proven superior in the CONCH comparison experiments we conducted.
> 2. We used all annotated patches to build the cache model.
>
>
> $\textbf{Q7:}$ Few-shot instances are randomly selected from the core set. Since only a few instances, e.g. 16, are selected from a very large instance pool and pathological patches (instances) often present heterogeneity, the final selected instance set could have a large variety and thus lead to unstable performances, calling into question the usability of the proposed method. This could also be observed from Fig 3 in the paper. So, the authors are encouraged to analyze and discuss the impact of randomly selected few-shot instances.
>
> $\textbf{R7:}$ We apologize for any inconvenience brought to you. In our constructed cache model, not only does it include a few labeled patches, but it also contains a large number of unlabeled patches, making our model relatively stable. Additionally, as shown in Figure 3, when the instance shot reaches 16, the variance becomes relatively small.
>
> $\textbf{Q8:}$ The original TCGA-RCC is not annotated at the pixel level. This work does a good job in terms of annotating the fine-grained region-of-interest of WSIs. However, if the annotation is not made public, this work would be difficult to follow and be very limited in research impact. The authors are encouraged to make their annotated dataset public.
>
> $\textbf{R8:}$ Thank you very much for your suggestions on our work. To promote progress and development in the community, we will release the relevant datasets available for academic research.

---

> ### Comment · Reviewer_x5Ho · 2024-08-09
> **Reply to the Authors' Rebuttal**
>
> Thanks for the authors' efforts and responses. After carefully reading the replies, I still have the following concerns:
> - **R2**: I am more concerned with the technical novelty since the proposed framework does incremental work to the existing Tip-Adapter. By the way, I do acknowledge the novelty of the two-level few-shot learning paradigm, as I mentioned in the Strengths.
> - **R4**: Thanks for the experimental results. These results could be helpful to justify the value of this work. However, I cannot agree with the authors' claims made in R4.
>   - "*Compared to CONCH, FAST significantly outperforms CONCH in the average bag-level classification AUC*". Is there any statistical test to verify the significance difference between the AUC of 0.9235 and 0.9141? Or, does this conclusion just come from a personal sense? I think in scientific research the conclusion given by the authors must be rigorous enough.
>   - "*while CONCH relies on 1.17 million pairs of pathology images based on CoCa, resulting in a significantly high training cost.*". CONCH is a foundational model for pathology, just like CLIP. Here, discussing its efficacy and comparing it with the proposed FAST is not appropriate, because I) foundation models generally rely on massive data and large-scale pretraining, and ii) FAST also stands on the shoulder of such foundation models like GPT and CLIP, right?
>   - Additionally, I mentioned a comparison with CONCH. It intends to encourage the authors to justify the significance and value of this work, not to question the validity of the experiments. Concretely, for example, if the foundational model, CONCH, could achieve an accuracy of 90% in zero-shot settings yet the FAST framework with CONCH only obtains 90.5% in few-shot settings, the improvement would be too marginal to demonstrate the value of the proposed few-shot FAST.
>   - "*Therefore, our method can be effectively combined with methods such as PLIP and CONCH to further enhance WSI classification performance.*". I failed to find the experiments on FAST + CONCH and see the improvements, so I think this claim, *i.e.*, it can further enhance WSI classification performance, may not be valid from my point of view.
> - **R6**: I cannot figure out the authors' implementation for Tip-Adapter in few-shot WSI classification, after carefully reading the authors' instructions. Could the authors please explain more? Thanks.
> - **R7**: The authors mention that the model is relatively stable because the cache model includes a few labeled patches and a large number of unlabeled patches. It seems not obvious to me, since I just don't understand the logic behind the cause and consequence provided by the authors.

---

> > ### Author Response · Authors · 2024-08-10
> > **Response to the Remaining Concerns from Reviewer x5Ho**
> >
> > Thank you very much for your rapid response, which is crucial for improving the quality of our manuscript.
> >
> > $\textbf{Response to R2:}$ First, we sincerely appreciate the reviewer’s recognition of our innovation in the dual-tier few-shot learning paradigm, which is meaningful for WSI classification. Additionally, when Tip-adapter is applied to WSI classification, it faces challenges such as the inability to fully utilize WSI data and the issue of huge size. For the former, directly adopting the methods from Tip-adapter to fully utilize WSI data would lead to instability in training, as mentioned in response to an earlier question. To address this, we proposed a cache model where both labels and features are learnable, with the labels of annotated patches being frozen while those of unannotated patches remain learnable. This effectively alleviates the issue of Tip-adapter being unable to fully utilize WSI data. For the latter, we introduced a core set construction method that effectively addresses the challenge of training on entire WSIs caused by their huge size. Overall, our model design is inspired by Tip-adapter, but it is not entirely identical to Tip-adapter.
> >
> > $\textbf{Response to R4:}$ We are very grateful for the problems pointed out by the reviewer, as they have been extremely helpful in improving the quality of our manuscript. The last three questions are closely related, so we will answer them together in response 2.
> > 1. In the final version of the paper, we will revise “Compared to CONCH, FAST significantly outperforms CONCH in the average bag-level classification AUC” to “Compared to CONCH, FAST outperformed CONCH by 0.0094 in the average bag-level classification AUC.”
> > 2. First, I would like to provide a brief explanation of PLIP and CONCH. PLIP is a version of CLIP fine-tuned on large-scale pathological data. Similar to CONCH, which is trained based on the large-scale vision-language model CoCa. Secondly, we apologize for any inconvenience caused. Below, we present the experimental results of our method combined with CONCH. As shown in the table A5, when the bag shot is 1 and the instance shot is 16, FAST-CONCH improves the instance-level AUC by 0.1227 and the bag-level AUC by 0.1485 compared to FAST-CLIP. When the bag shot is 16 and the instance shot is 16, FAST-CONCH improves the instance-level AUC by 0.0615 and the bag-level AUC by 0.1373 compared to FAST-CLIP. CONCH achieved a bag-level classification AUC of 0.7113 on the CAMELYON16 dataset. In comparison, FAST-CONCH improved this by 0.2457.
> >
> > $\textbf{Table A5: Results of using CONCH on CAMELYON16 dataset.}$
> > | Bag Shot | Instance Shot | Methods | Instance-level AUC | Bag-level AUC |
> > | :---: | :---: | :---: | :---: | :---: |
> > | 0 | 0 | CONCH | $0.8929$ | $0.7113$ |
> > | 1 | 16 | FAST-CLIP | $0.8400 \pm 0.0335$ | $0.6933 \pm 0.0846$ |
> > | 1 | 16 | FAST-CONCH | $0.9627 \pm 0.0132$ | $0.8418 \pm 0.0734$ |
> > | 2 | 16 | FAST-CLIP | $0.8584 \pm 0.0380$ | $0.7595 \pm 0.0391$ |
> > | 2 | 16 | FAST-CONCH | $0.9667 \pm 0.0115$ | $0.8399 \pm 0.0556$ |
> > | 4 | 16 | FAST-CLIP | $0.8864 \pm 0.0563$ | $0.7359 \pm 0.0853$ |
> > | 4 | 16 | FAST-CONCH | $0.9763 \pm 0.0036$ | $0.9326 \pm 0.0175$ |
> > | 8 | 16 | FAST-CLIP | $0.9060 \pm 0.0074$ | $0.7742 \pm 0.0249$ |  |
> > | 8 | 16 | FAST-CONCH | $0.9792 \pm 0.0024$ | $0.9507 \pm 0.0058$ |
> > | 16 | 16 | FAST-CLIP | $0.9151 \pm 0.0200$ | $0.8197 \pm 0.0474$ |
> > | 16 | 16 | FAST-CONCH | $0.9766 \pm 0.0036$ | $0.9570 \pm 0.0053$ |
> >
> > $\textbf{Response to R6:}$ In WSI classification, we followed the settings from the original Tip-adapter paper. For aspects that were not directly applicable to WSI classification task, we made the following adjustments: 1. We designed a specific set of text prompts customized for WSI classification task. 2. We used all annotated patches to construct the cache model. Apart from these differences, the settings are consistent with those in the Tip-adapter paper.
> >
> > $\textbf{Response to R7:}$
> > We sincerely apologize for any confusion we may have caused. We randomly selected a small number of instances from the core set for annotation. The remaining instances in the core set were not annotated but instead were all used in the training of the cached model. During training, the unannotated patches gradually learn the labels used for classification from the annotated patches. Since all patches in the core set are eventually used for model training, our model is relatively stable. The variance observed by the reviewer in Figure 3 comes from the random selection of bags. When the number of bags is 1 or 2, the variance is indeed large. We believe this is reasonable, as it is challenging to fit all the data with just one WSI. When the number of bags is greater than or equal to 4, the variance decreases considerably. Therefore, in practical applications, we recommend selecting 4 bags or more.

---

> ### Comment · Reviewer_x5Ho · 2024-08-12
>
> I would like to thank the authors for their efforts. Most of the time during the rebuttal, I feel inefficient communication in reading the author's responses, as most of these responses often fail to capture the true meaning of my questions and lead to undesirable QA.
>
>
> Yet, overall, most of my concerns have been resolved. Thanks for the authors' experiments provided in the rebuttal. I believe the proposed framework could facilitate the study of few-shot WSI analysis. In view of these, I am happy to increase my score. The authors are encouraged to include the important suggestions & questions into the final version of the paper.

---

> > ### Author Response · Authors · 2024-08-13
> >
> > Thank you very much for your suggestions, which have been extremely helpful in improving the quality of our manuscript. We also sincerely appreciate your recognition of our work and the higher score. We will incorporate the aforementioned content in the camera-ready version of the paper.

---

### Official Review · Reviewer_hypM · 2024-07-03

**Soundness:** 3
**Presentation:** 3
**Contribution:** 4
**Rating:** 8
**Confidence:** 5

**Summary:**

To address the challenges of expensive fine-grained annotation and data scarcity encountered in the clinical application of deep learning-based WSI classification methods, this paper proposes a novel and efficient dual-tier few-shot learning paradigm named FAST. Under this new paradigm, the authors introduce a dual-level annotation strategy that includes bag-level few-shot and instance-level few-shot, modeling the WSI classification problem as a new few-shot classification problem. Building on this, the authors further propose a classification framework composed of a learnable image cache branch and a CLIP prior knowledge branch, fully leveraging the value of the limited data and labels. Extensive experimental results show significant improvement over other few shot methods in both binary and multi-class classification tasks. Interestingly, the proposed method FAST achieves performance close to fully supervised methods with only 0.22% of the annotation cost. This showcases its efficiency and great potential for practical applications.

**Strengths:**

1. The paper is well written and easy to read. The authors intuitively demonstrate their methods and contributions through numerous figures and tables. Extensive comparative and ablation experiments illustrate the efficiency and generality of the proposed method. To ensure fairness and prevent randomness, the authors conducted multiple random experiments in their study. The results show significantly better performance in both bag-level and instance-level classification compared to other methods.
2. The proposed dual-level WSI annotation strategy is a highly innovative and suitable method for WSI data annotation. It addresses the issues of single-level annotation and provides patch-level supervisory information at a cost close to slide-level annotation. Compared to fully supervised methods, the proposed method has significantly lower annotation costs, astonishingly reaching as low as one-thousandth or even one-ten-thousandth.
3. This paper is inspired by Tip-adapter and proposes a learnable cache branch where both labels and image features are learnable. The final classification framework further integrates a CLIP prior knowledge branch incorporating GPT-4V. Comparative experiments show that this method achieves performance close to fully supervised methods with only 0.2% of the data annotation cost, which is an exciting advancement. Ablation experiments also demonstrate the importance of the proposed components.

**Weaknesses:**

1. The function ϕ(∙) in Figure 2 is not mentioned or explained in the paper, which may confuse readers.
2. The authors conducted extensive comparisons in terms of accuracy and annotation cost but lacked analysis of time and memory consumption.
3. In section 3.2, the authors mention obtaining the optimal fusion weight α through grid search but lack specific details in the paper.

**Questions:**

Will the authors open source the relevant code and all model weights for this project?
For other issues, please refer to the weaknesses.

**Limitations:**

In the conclusion section, the authors acknowledge the limitations of their proposed method. I agree that such limitations exist and look forward to future work.

---

> ### Author Rebuttal · Authors · 2024-08-05
>
> $\textbf{Q1:}$ The function $\phi(\cdot)$ in Figure 2 is not mentioned or explained in the paper, which may confuse readers.
>
> $\textbf{R1:}$ We apologize for any inconvenience brought to you. We have added a description of the function \phi(\cdot) in Section 3.2, revising “The retrieval result is $(\dot{Q}\dot{K}^T)\dot{V}$ ” to “ The retrieval result is $\phi(\dot{Q} \dot{K}^T)\dot{V}$ , where $\phi(\cdot) = \text{softmax}(\cdot)$ . ”
>
> $\textbf{Q2:}$ The authors conducted extensive comparisons in terms of accuracy and annotation cost but lacked analysis of time and memory consumption.
>
> $\textbf{R2:}$ Thanks for your great suggestion on improving the quality of our manuscript. We conducted experiments on training time and memory consumption for the scenario with an instance shot of 16 and a bag shot of 16 using an NVIDIA RTX 3090. The results are shown in the Table A3. Our method achieves performance close to fully supervised methods with a training time of only 0.21 hours and a memory usage of just 5.12 GB. In comparison, training pathology large models like CONCH requires 8 NVIDIA A100 GPUs, highlighting a significant advantage of our method.
>
> $\textbf{Table A3: analysis of time and memory consumption}$
> | Metric | Time (h) | Memory (GB) |
> | :---: | :---: | :---: |
> | FAST | 0.21 | 5.12 |
>
> $\textbf{Q3:}$ In section 3.2, the authors mention obtaining the optimal fusion weight α through grid search but lack specific details in the paper.
>
> $\textbf{R3:}$ Thanks for your great suggestion on improving the quality of our manuscript. We have added the following description in Section 3.2. We divide the fusion weight \alpha into equal intervals with a step size of 100, then sequentially calculate the classification accuracy for each fusion ratio, and finally select the fusion weight that yields the highest classification accuracy as the fusion weight \alpha for this task.
>
>
> $\textbf{Q4:}$ Will the authors open source the relevant code and all model weights for this project?
>
> $\textbf{R4:}$ Thank you very much for your recognition of our work. We will open-source all related code and model weights to promote further development in WSI classification research.

---

> > ### Comment · Reviewer_hypM · 2024-08-10
> > **Thank you for your rebuttal, which clearly addressed my concerns.**
> >
> > Thank you for your rebuttal, which clearly addressed my concerns. I have read other reviewers' comments and the author's rebuttal, particularly the comparison with vision-language-based models in computational pathology. The experimental results and the authors’ responses clearly and effectively demonstrate the contribution of this paper. Overall, the authors' rebuttal resolves my concerns, and their answers to other reviewers' questions also seem reasonable to me. I think this paper is highly valuable for advancing computational pathology. Thus, I increase my rating to strong accept.

---

> > > ### Author Response · Authors · 2024-08-13
> > >
> > > Thank you very much for your suggestions, which have been extremely helpful in improving the quality of our manuscript. We also sincerely appreciate your recognition of our work and the higher score. We will incorporate the aforementioned content in the camera-ready version of the paper.

---

### Official Review · Reviewer_HxPT · 2024-07-08

**Soundness:** 3
**Presentation:** 3
**Contribution:** 3
**Rating:** 6
**Confidence:** 3

**Summary:**

In this article, the authors propose a novel few-shot learning paradigm for WSI classification. This paradigm is based on two branches: the first is a learnable cache model that utilizes both labeled and unlabeled instance data, and the second, the Prior Branch, leverages the prior knowledge of a pre-trained CLIP model. By combining these two branches, efficient few-shot learning is achieved, and extensive experiments have been conducted on the CAMELYON16 dataset and the TCGA-RENAL dataset.

**Strengths:**

1.	Few-shot learning is inherently important in the field of WSI classification. The authors have proposed a new few-shot learning paradigm tailored for WSI classification and have achieved notable results.
2.	The experiments on few-shot learning are quite comprehensive, thoroughly comparing the effects of different numbers of instances and bags.

**Weaknesses:**

1. The study lacks experiments with V-L models specific to the pathology field. Since CLIP is not originally based on pathology images, the authors should include comparisons using PLIP [1] and CONCH [2].
2. The few-shot learning method proposed by the authors operates at both the instance-level and bag-level. Therefore, the comparative methods should include both instance-based methods and bag-level methods (multi-instance learning). However, the fully supervised methods chosen for comparison are only instance-based. The authors should supplement their comparisons with bag-level methods based on multi-instance learning, such as R2T [3].
3. Comparing the third and fourth rows in Table 3 of the paper reveals that adding the Prior Branch on top of existing components brings almost no improvement. However, it requires first processing through GPT and then the Text-encoder, significantly increasing the cost without enhancing performance.

[1] PLIP: A visual–language foundation model for pathology image analysis using medical Twitter. Nature Medicine 2023
[2] CONCH：A Vision-Language Foundation Model for Computational Pathology. Nature Medicine 2024
[3] Feature Re-Embedding: Towards Foundation Model-Level Performance in Computational Pathology. CVPR 2024

**Questions:**

What the difference between the proposed work and previous few-shot WSI classification methods like [4]?

[4] The rise of ai language pathologists: Exploring two-level prompt learning for few-shot weakly-supervised whole slide image classification. NeurIPS 2023

**Limitations:**

The authors only tested up to 16 instances and bags, but there is still a significant performance improvement from 8 to 16. I am curious at what data ratio in few-shot learning the model will begin to overfit.

---

> ### Author Rebuttal · Authors · 2024-08-05
>
> $\textbf{Q1:}$ The study lacks experiments with V-L models specific to the pathology field. Since CLIP is not originally based on pathology images, the authors should include comparisons using PLIP [1] and CONCH [2].
>
> $\textbf{R1:}$ Thanks for your great suggestion on improving the quality of our manuscript. Due to the huge size of WSIs, extracting features for the entire dataset using foundational models such as CLIP, PLIP, and CONCH requires a significant amount of time. We have not yet completed the extraction of features encoded using PLIP and CONCH. Additionally, many function wrappers in PLIP and CONCH are different from those in CLIP. Therefore, we need more time to build models using PLIP and CONCH. Once we obtain the latest experimental results, we will include these results in our subsequent replies. We have now completed the extraction of all features for the test set and have conducted some zero-shot classification experiments. Please refer to Table A1 and Table A2 in the PDF file for the experimental results.
>
> $\textbf{Q2:}$ The few-shot learning method proposed by the authors operates at both the instance-level and bag-level. Therefore, the comparative methods should include both instance-based methods and bag-level methods (multi-instance learning). However, the fully supervised methods chosen for comparison are only instance-based. The authors should supplement their comparisons with bag-level methods based on multi-instance learning, such as R2T [3].
>
> $\textbf{R2:}$ Thank you very much for your valuable suggestion. We did not compare bag-level methods based on multi-instance learning for the following reasons. First, instance-level fully supervised methods represent the upper bound of supervised learning classification results. Bag-level methods generally perform worse than instance-level fully supervised methods due to the lack of precise fine-grained labels. Second, we found it challenging to accurately reproduce the results of R2T within the limited time available. For these reasons, we did not include this comparison experiment. However, we have added an explanation in Section 4.2, “Comparing Methods and Evaluation Metrics,” about why we did not compare with “bag-level methods,” and we have also included references to relevant methods in the related work section. The specific content added is as follows. “Instance-level fully supervised methods represent the upper bound of supervised learning. Bag-level weakly supervised multi-instance learning methods, such as R2T, generally perform worse than instance-level fully supervised methods due to the lack of fine-grained labels. Therefore, this paper does not directly compare with multi-instance learning methods.”
>
> $\textbf{Q3:}$ Comparing the third and fourth rows in Table 3 of the paper reveals that adding the Prior Branch on top of existing components brings almost no improvement. However, it requires first processing through GPT and then the Text-encoder, significantly increasing the cost without enhancing performance.
>
> $\textbf{R3:}$ We apologize for any inconvenience brought to you. If we only compare the third and fourth rows of Table 3, it might indeed seem that way. However, we found that these results in Table 3 are due to the experiments being conducted under the 16-bag shot setting. To fully demonstrate the role of the prior branch, we conducted further experiments, and the related results and analysis can be found in the main text, Figure 4, and line 288. For convenience, we have included some key conclusions here: “When there are only 1 or 2 bags, the instance classification results of the prior branch are significantly higher than those of the cache branch. The instance and bag classification results that combine both the cache and prior branches also surpass those of using each branch separately, indicating that the prior branch performs better in extreme samples, and the information learned by the prior branch and the cache branch is complementary. Therefore, in extreme few-shot scenarios, FAST is dominated by the prior branch, but as the sample size gradually increases, FAST is dominated by the image branch.”
>
> $\textbf{Q4:}$ What the difference between the proposed work and previous few-shot WSI classification methods like [4]? [4] The rise of ai language pathologists: Exploring two-level prompt learning for few-shot weakly-supervised whole slide image classification. NeurIPS 2023
>
> $\textbf{R4:}$ Our method differs significantly from Top in the following ways.
> 1. Different Scenarios: Top is a few-shot learning method under slide-level labels. While it also reduces annotation costs, it lacks precise patch-level label information. In contrast, we propose a dual few-shot learning scenario tailored for WSIs, which not only provides patch-level label information but also significantly reduces annotation costs.
> 2. Different Technical Approaches: The focus of Top’s research is on designing better text prompt strategies to serve CLIP. In contrast, our method introduces new approaches from both the image cache branch and the text prior branch to fully utilize existing annotation information.
>
> $\textbf{Q5:}$ The authors only tested up to 16 instances and bags, but there is still a significant performance improvement from 8 to 16. I am curious at what data ratio in few-shot learning the model will begin to overfit.
>
> $\textbf{R5:}$ We conducted experiments with 64 instances in the supplementary materials, and the results are shown in Figure 6. From Figure 6, it can be observed that as the instance shot increases, the classification accuracy also gradually improves. When the number of shots reaches 64, the rate of accuracy increase slows down. Therefore, considering both training costs and performance benefits, we recommend using an instance shot of 16 or 64 in practical applications.

---

> ### Author Response · Authors · 2024-08-09
> **Experiments with V-L models specific to the pathology field**
>
> Thank you very much for your valuable suggestions. We have implemented FAST using the vision-language model CONCH from the pathology field and conducted experiments on the CAMELYON16 dataset. The experimental results are shown in Table A4.  We define the method using CLIP as the feature extractor as FAST-CLIP, and the method using CONCH as the feature extractor as FAST-CONCH. As shown in Table A4, compared to using CLIP as the feature extractor, using CONCH significantly improves the classification performance of FAST. Notably, the bag-level classification AUC can reach 0.957. This indicates that our method not only integrates well with V-L models like CLIP in natural images for WSI classification but also enhances the classification performance of V-L models in pathology.
>
> $\textbf{Table A4: Results of using CONCH on CAMELYON16 dataset.}$
> | Bag Shot | Instance Shot | Methods | Instance-level AUC | Bag-level AUC |
> | :---: | :---: | :---: | :---: | :---: |
> | 1 | 16 | FAST-CLIP | $0.8400 \pm 0.0335$ | $0.6933 \pm 0.0846$ |
> | 1 | 16 | FAST-CONCH | $0.9627 \pm 0.0132$ | $0.8418 \pm 0.0734$ |
> | 2 | 16 | FAST-CLIP | $0.8584 \pm 0.0380$ | $0.7595 \pm 0.0391$ |
> | 2 | 16 | FAST-CONCH | $0.9667 \pm 0.0115$ | $0.8399 \pm 0.0556$ |
> | 4 | 16 | FAST-CLIP | $0.8864 \pm 0.0563$ | $0.7359 \pm 0.0853$ |
> | 4 | 16 | FAST-CONCH | $0.9763 \pm 0.0036$ | $0.9326 \pm 0.0175$ |
> | 8 | 16 | FAST-CLIP | $0.9060 \pm 0.0074$ | $0.7742 \pm 0.0249$ |  |
> | 8 | 16 | FAST-CONCH | $0.9792 \pm 0.0024$ | $0.9507 \pm 0.0058$ |
> | 16 | 16 | FAST-CLIP | $0.9151 \pm 0.0200$ | $0.8197 \pm 0.0474$ |
> | 16 | 16 | FAST-CONCH | $0.9766 \pm 0.0036$ | $0.9570 \pm 0.0053$ |

---

> > ### Comment · Reviewer_HxPT · 2024-08-14
> >
> > Thanks for the authors responce and further experiments. My questions are addressed, and I will rise the score.

---

> > > ### Author Response · Authors · 2024-08-14
> > >
> > > Thank you very much for your suggestions, which have greatly helped improve the quality of our manuscript. We sincerely appreciate your recognition of our work and the higher score. We will include the aforementioned content in the camera-ready version of the paper.

---

### Official Review · Reviewer_Bx9c · 2024-07-10

**Soundness:** 4
**Presentation:** 4
**Contribution:** 3
**Rating:** 7
**Confidence:** 5

**Summary:**

This paper investigates the issue of Whole Slide Images (WSI) classification, a study with practical value. It proposes a new working paradigm that is an improvement based on Tip-Adapter. Theoretically, this new paradigm can effectively address the problem and has strong scalability.

**Strengths:**

This study has practical significance, and the proposed method demonstrates strong scalability. The paper is clearly written and easy to understand, with comprehensive experiments.

**Weaknesses:**

1. This paper lacks some important related work. The proposed method is based on the Tip-Adapter. While, there are many improvements based on Tip-Adapter, such as [1-4].  I think the experiments should include comparisons with these related methods, or at the very least, mention and briefly analyze them.
[1] Collaborative Consortium of Foundation Models for Open-World Few-Shot Learning. AAAI, 2024.
[2] Prompt, Generate, then Cache: Cascade of Foundation Models makes Strong Few-shot Learners. CVPR, 2023.
[3] DeIL: Direct-and-Inverse CLIP for Open-World Few-Shot Learning. CVPR, 2024.
[4] Not All Features Matter: Enhancing Few-shot CLIP with Adaptive Prior Refinement. ICCV, 2023.

2. In Figure 1, the blue and red boxes should be explained. Additionally, if space permits, I suggest that in future work, the authors could add more detailed descriptions in the caption. By this way, readers can understand the general idea of the method just by looking at the figure and caption, without having to spend effort finding the corresponding description in the main text.

3. For instance-shot, only the results of 16-shot are shown, without the results of 1-shot, 2-shot, etc.
4. Line 160 seems to have a typo; it should be y_1L instead of y_1V.

**Questions:**

see weakness

**Limitations:**

authors didn't present the limitation.

---

> ### Author Rebuttal · Authors · 2024-08-05
>
> $\textbf{Q1:}$ This paper lacks some important related work. The proposed method is based on the Tip-Adapter. While, there are many improvements based on Tip-Adapter, such as [1-4]. I think the experiments should include comparisons with these related methods, or at the very least, mention and briefly analyze them.
>
> $\textbf{R1:}$ Thanks for your great suggestion on improving the quality of our manuscript. our method is orthogonal to these four studies, and they do not conflict with each other. Additionally, the four studies mentioned above mainly focus on natural images and would face similar challenges as Tip-Adapter when directly applied to WSIs. Combining our method with these four methods has great potential to further improve the WSIs classification accuracy. For these reasons, we did not compare them directly through experiments, but we analyzed and summarized these four important studies in the related work section. The specific content added in the related work section is as follows.
> In the field of natural images, many subsequent works based on Tip-Adapter have also made significant contributions to the development of foundation model adaptation. For example, CaFo [2] effectively combines the different prior knowledge of various pre-trained models by cascading multiple foundation models. CO3 [1] goes a step further by considering both general and open-world scenarios, designing a text-guided fusion adapter to reduce the impact of noisy labels. Similarly, for open-world few-shot learning, DeIL [3] proposes filtering out less probable categories through inverse probability prediction, significantly improving performance. APE [4] proposes an adaptive prior refinement method that significantly enhances computational efficiency while ensuring high-precision classification performance. Due to the huge size and the lack of pixel-level annotations, these methods cannot effectively solve the classification problem of WSIs.
>
>
> $\textbf{Q2:}$ In Figure 1, the blue and red boxes should be explained. Additionally, if space permits, I suggest that in future work, the authors could add more detailed descriptions in the caption. By this way, readers can understand the general idea of the method just by looking at the figure and caption, without having to spend effort finding the corresponding description in the main text.
>
> $\textbf{R2:}$ Thanks very much for pointing out the problem. We have added the following description below Figure 1. Figure 1: Different few-shot learning paradigms for WSI classification. (a) The instance few-shot method divides all WSIs into a series of patches, then selects a few samples at the patch level and annotates them at the patch level. The red box represents positive samples, and the blue box represents negative samples. (b) The bag few-shot method directly selects a few WSIs at the slide level and annotates them weakly at the slide level. (c) Our method first selects a few WSIs at the slide level, then annotates a few patches for each selected WSI. Compared to (a) and (b), our method significantly reduces annotation costs while providing patch-level supervision information.
>
>
> $\textbf{Q3:}$  For instance-shot, only the results of 16-shot are shown, without the results of 1-shot, 2-shot, etc.
>
> $\textbf{R3:}$ We apologize for any inconvenience brought to you. Due to space limitations in the main text, we did not present the experimental results for different shot settings. Instead, we included these results on page 22 of the supplementary materials. In Figure 6, we show the results for 4-shot, 16-shot, and 64-shot settings. From Figure 6, it can be observed that the classification accuracy gradually increases with the number of shots. When the number of shots reaches 64, the accuracy increase nearly converges. Therefore, we recommend using 16-shot or 64-shot in practical applications. We have also added experimental results for the 1-shot and 2-shot settings. The results are shown in Figure A1, which we have uploaded in the PDF file. When the number of shots decreases to 1-shot or 2-shot, the accuracy decreases, with the lower limit being the result of Zero-shot CLIP.
>
>
> $\textbf{Q4:}$ Line 160 seems to have a typo; it should be $y_{1,L}$ instead of $y_{1,V}$.
>
> $\textbf{R4:}$ Thanks very much for pointing out the problem. The correct term here should indeed be $y_{1,L}$ . We have corrected this typo in the manuscript.

---

> > ### Comment · Reviewer_Bx9c · 2024-08-11
> > **Response to the Authors**
> >
> > Thanks for the authors' responses. The author has resolved my questions and I agree to accept this paper.

---

> > > ### Author Response · Authors · 2024-08-13
> > >
> > > Thank you very much for your suggestions, which have been extremely helpful in improving the quality of our manuscript. We also sincerely appreciate your recognition of our work and the higher score. We will incorporate the aforementioned content in the camera-ready version of the paper.

---

### Author Rebuttal · Authors · 2024-08-05

We sincerely thank all reviewers for your valuable comments. These comments have greatly helped improve the quality of our manuscript. Next, we will reply to the questions raised by each reviewer individually. The figures and tables mentioned in our replies have all been uploaded in a single PDF file.

---

### Decision · Program_Chairs · 2024-09-25

**Decision:**

Accept (poster)

**Comment:**

The paper studies few-shot learning for whole slide classification, and presents a method that can harness pre-trained language and vision-language models. After the rebuttal and discussion, the reviewers unanimously recommend acceptance of the paper. They recognize the significant practical implications of the proposed few-shot learning paradigm, and commend the strong empirical results. The proposed few-shot method demonstrates an impressive ability to match the performance of fully supervised learning while significantly reducing annotation costs, which all reviewers find particularly compelling.